# Plant Proteins and Exercise: What Role Can Plant Proteins Have in Promoting Adaptations to Exercise?

**DOI:** 10.3390/nu13061962

**Published:** 2021-06-07

**Authors:** Chad M. Kerksick, Andrew Jagim, Anthony Hagele, Ralf Jäger

**Affiliations:** 1Exercise and Performance Nutrition Laboratory, School of Health Sciences, College of Science, Technology, and Health, Lindenwood University, St. Charles, MO 63301, USA; amh524@lindenwood.edu; 2Sports Medicine, Mayo Clinic Health System, La Crosse, WI 54601, USA; jagim.andrew@mayo.edu; 3Increnovo, LLC, Milwaukee, WI 53202, USA; ralf.jaeger@increnovo.com

**Keywords:** plants, complete, incomplete, protein, exercise, fat-free mass, training adaptations, performance, recovery

## Abstract

Adequate dietary protein is important for many aspects of health with current evidence suggesting that exercising individuals need greater amounts of protein. When assessing protein quality, animal sources of protein routinely rank amongst the highest in quality, largely due to the higher levels of essential amino acids they possess in addition to exhibiting more favorable levels of digestibility and absorption patterns of the amino acids. In recent years, the inclusion of plant protein sources in the diet has grown and evidence continues to accumulate on the comparison of various plant protein sources and animal protein sources in their ability to stimulate muscle protein synthesis (MPS), heighten exercise training adaptations, and facilitate recovery from exercise. Without question, the most robust changes in MPS come from efficacious doses of a whey protein isolate, but several studies have highlighted the successful ability of different plant sources to significantly elevate resting rates of MPS. In terms of facilitating prolonged adaptations to exercise training, multiple studies have indicated that a dose of plant protein that offers enough essential amino acids, especially leucine, consumed over 8–12 weeks can stimulate similar adaptations as seen with animal protein sources. More research is needed to see if longer supplementation periods maintain equivalence between the protein sources. Several practices exist whereby the anabolic potential of a plant protein source can be improved and generally, more research is needed to best understand which practice (if any) offers notable advantages. In conclusion, as one considers the favorable health implications of increasing plant intake as well as environmental sustainability, the interest in consuming more plant proteins will continue to be present. The evidence base for plant proteins in exercising individuals has seen impressive growth with many of these findings now indicating that consumption of a plant protein source in an efficacious dose (typically larger than an animal protein) can instigate similar and favorable changes in amino acid update, MPS rates, and exercise training adaptations such as strength and body composition as well as recovery.

## 1. Introduction

### 1.1. Muscle Protein Metabolism and Skeletal Muscle

The maintenance of and optimizing the accretion of skeletal muscle mass are critical outcomes for athletic-minded individuals, whether the goal is increased performance, improved muscularity, or enhanced recovery. Furthermore, while skeletal muscle mass accretion is often a goal of active individuals, there are direct clinical applications and benefits for the general public as well, especially for aging adults. Skeletal muscle is regulated through a near-continual ebb and flow between rates of muscle protein synthesis (MPS) and breakdown [1]. Muscle mass loss occurs during a net negative balance (breakdown > synthesis) while muscle gain occurs when synthesis rates outweigh breakdown. Rates of MPS and muscle protein breakdown are highly sensitive to physical activity and dietary intake, namely protein and essential amino acid intake [2], with evidence available indicating that rates of MPS are more sensitive to changes in exercise status and dietary intake [3]. As a result, observed changes in MPS rates are viewed to be primarily responsible for the changes in muscle mass in response to exercise and nutrition experienced over time [4].

### 1.2. The Importance of Added Protein to Optimize Exercise Training Adaptations

Supplementing the diet with added protein beyond the recommended dietary allowance (RDA) has long been a well-supported tactic for exercising athletes to optimize exercise training adaptations. In this respect, multiple review articles and position stands have advocated for a greater intake of dietary protein to support increased physical training volumes, heighten exercise training adaptations, and promote health and recovery [5,6,7,8,9]. Previously, Cermak and colleagues [10] completed a meta-analysis of studies that employed some form of protein supplementation while completing resistance training. Results from this analysis included data from over 680 subjects and concluded that protein supplementation led to a significantly greater increase in fat-free mass (mean difference: 0.69 kg, 95% CI: 0.47–0.91 kg, *p* < 0.001) and maximal lower-body strength (mean difference: 13.5 kg, 95% CI: 6.4–20.7 kg, *p* < 0.005) when compared to a placebo. These results were extended by Morton and investigators [7] who used a meta-analysis and meta-regression approach to establish the efficacy of protein supplementation while also identifying the minimum amount of daily protein needed to maximize efficacy. In this study, 49 studies were included that represented 1863 participants and the authors reported that protein supplementation was responsible for significant increases in strength (1 RM), fat-free mass, and muscle cross-sectional area. Moreover, results from this study highlighted that a daily protein intake beyond 1.62 g/kg/day offered no further impact in facilitating improvements in fat-free mass. It is important to note, this is well above (~2×) the RDA for protein, indicating that active individuals benefit from consuming greater amounts of protein. Whether or not higher amounts facilitate improvements in other outcomes such as strength, recovery, mitigation of fat-free mass loss seen while dieting was not identified in their analysis. Notably, this amount of protein is consistent with the protein recommendation set forth by Jäger and colleagues [5] in the position stand published by the International Society of Sports Nutrition as well as the position stand endorsed by the American College of Sports Medicine, Dietitians of Canada and Academy of Nutrition and Dietetics (the former American Dietetic Association) [11]. Dietary proteins are well known to serve as the primary supplier of amino acids that can be used as building blocks to make larger proteins, such as those produced during MPS. Previous studies have highlighted the importance of the essential amino acids [12,13] at stimulating rates of MPS. In addition, extensive research continues to explore the role of leucine in its ability to stimulate the initiation of protein translation [14,15]. All things considered, exercising individuals require greater amounts of dietary protein to support their training needs, which creates a need for these individuals to purposefully include various sources of protein that deliver optimal amounts of the essential amino acids.

### 1.3. The Case for Plant Proteins

Many sources of protein are available for consumption in the human diet. For years, heavy emphasis was placed on consuming complete protein sources, or any protein source that provides all of the essential amino acids in both the needed amount and in adequate proportion to support cellular needs across the body as well as production of nonessential amino acids [8]. Consequently, great focus has been placed on consuming animal protein sources, namely because of their high amino acid contents and favorable protein quality ratings [16]. At the same time, plant proteins were deemed inferior for these outcomes and not until recently has interest in plant proteins begun to accelerate. Several reasons are commonly associated with consuming greater amounts of plant proteins. Most commonly, plant-based diets are routinely linked with reductions in the risk of developing cancers, type 2 diabetes, and cardiovascular diseases [17]. In addition, many plant protein advocates highlight a greater level of economic sustainability than what is observed with diets that are predominantly animal protein. Finally, approximately 60% of dietary proteins consumed worldwide come from plant sources with an estimated 4 billion people across the globe consuming a primarily plant-based diet [18]. While such health considerations are unquestionably important, the aim of this review will center upon the implications of plant protein consumption and plant-based diets on outcomes linked to exercise performance, associated exercise training adaptations, and recovery.

### 1.4. Quality Considerations for Both Animal and Plant-Based Protein

Many factors contribute to the anabolic potential of a protein source, which often include the amounts of total amino acids, essential amino acids, and branched-chain amino acids, respectively in addition to the protein’s digestibility, digestion rate, and kinetics observed during absorption. In this respect, dietary protein quality is commonly assessed based upon the essential amino acid composition provided by the protein source as it relates to human needs, against the ability of the protein to be digested, absorbed, and assimilated by various tissues in the body [19]. Several approaches have been used to assess protein quality including biological value, net protein utilization, and protein digestibility corrected amino acid scores (PDCAAS) [20], while digestible indispensable amino acid score (DIAAS) have been more recently proposed. As seen in an excellent review by Berrazaga et al. [16], biological values for common plant sources range from 56–74 while ranges of 77–104 are reported for various animal sources on theoretical 100-point scales. A similar dichotomy is observed for net protein utilization values, whereby plant sources range from 53–67 while animal sources range from 73–94 on a 100-point scale. One of the most commonly used quality comparators is that of Protein Digestibility Corrected Amino Acid Scores (PDCAAS) [21]. When using this approach, a score of 100 suggests that after considering its fecal digestibility, a given protein source can fully deliver all of the essential amino acids required by the body. In this respect, animal protein sources such as casein, whey, milk, and eggs all have scores of 100 while red meat has a score of 92. In contrast, all other common sources of plant proteins have PDCAAS values below 100 (commonly reported range of 45–75 per Barrazaga et al. [16]), with soy protein being the only exception, which has a score of 100. Similarly, if the DIAAS approach is used to assess protein quality, a similar trend is observed in that animal sources are commonly above 100 while nearly all plant sources are below 100. In this respect, Gorissen et al. [22] compared the amino acid contents of various sources of plant-based isolates against common sources of animal proteins and human skeletal muscle samples. Again, it was illustrated that many plant protein sources have inadequate amounts of certain amino acids (e.g., lysine, methionine) while also consistently having lower amounts of the essential and branched-chain amino acids, particularly when compared to animal protein sources as well as the amino acid content found in human skeletal muscle. To further reiterate this point, van Vliet and colleagues [23] have indicated previously that essential amino acid composition of a protein source was predictive of skeletal muscle’s anabolic potential and that all essential amino acids should be present in optimal amounts. For these reasons, higher quality sources of protein (at least when viewed in the context of amino acid profiles) should serve as more effective protein sources in terms of anabolic potential and its innate ability to facilitate skeletal muscle accretion and promote other desired adaptations. Finally, leucine content of a protein source continues to get interest for its role in initiating the translation of muscle proteins [14,15]. Towards this end, a general acceptance has suggested the leucine content of a protein source functions as a vital and reliable predictor of MPS rates. When leucine contents are compared across different protein sources, whey protein is the highest (~12–14%) [22], which aligns with whey protein’s superior ability to stimulate MPS rates when compared to isocaloric and isonitrogenous amounts of other protein sources [24]. Moreover, animal protein sources generally have higher amounts of leucine (8–9% for non-dairy animal sources) and >10% for dairy protein sources while plant sources routinely have a leucine content of 6–8% [22,23].

Beyond amino acid content, digestibility and absorption kinetics can also influence the value of a protein. In terms of digestibility, it is well documented that the digestibility of many sources of plants is much lower (45–80%) than what is observed with various animal proteins (>90%) [25]. While somewhat beyond the scope of this review, the observed differences in digestibility are largely thought to be due to structural differences that exist within the actual protein molecule found in many plant and animal proteins. For example, many sources of plants have compounds (i.e., anti-nutritional factors such as phytic acid, protease inhibitors, tannins, etc.) that compromise their digestibility. Another key factor related to the impact of consuming different sources of protein is the absorption of amino acids in plasma followed by the utilization rates exhibited by various proteins. In this respect, several studies have illustrated divergent utilization rates when comparing animal to plant sources of protein. For example, the classic work of Boirie [26,27] and Dangin [28,29] clearly demonstrated different absorption and utilization rates for two milk proteins, whey and casein. Moreover, the observed differences in rates of muscle protein metabolism have been shown to be inextricably linked to differences in utilization rates whereby whey absorbs faster and robustly stimulates rates of MPS while casein absorbs at a slower rate and consequently functions more to attenuate protein breakdown. When considering differences observed for various plant proteins, previous work has shown that soy ingestion is absorbed at a slower rate than what is observed from whey [24,30], which helps to explain the lower rates of myofibrillar protein synthesis observed by Yang and colleagues [4] after graded doses (0–40 g) of soy isolate at rest and after exercise in elderly men. While rates of myofibrillar protein synthesis were observed to increase with an increase in the dose of soy protein, the observed rates were less than what had been previously observed with equivalent doses of whey [24]. Additional research involving wheat proteins demonstrated them to have higher deamination rates when compared to milk proteins (25% vs. 16%, respectively) [31,32,33]. These differences are important as they are thought to be directly related to the lower observed net protein utilization rates between wheat (66%) when compared to milk (80%) proteins. Furthermore, other studies have illustrated a greater degradation of amino acids from soy protein when compared to degradation rates observed for casein and whey [24,30,34,35]. Towards this end, measured nitrogen losses (either via deamination or intestinal loss) and splanchnic nitrogen retention are higher when plant proteins are consumed when compared to ingestion of animal proteins. In effect, these outcomes illustrate that the availability of amino acids to peripheral tissues and locations from plant proteins is lower than that of animal protein [36,37] and these differences are thought to be key drivers to the post-prandial protein synthetic response observed in various tissues. Importantly, the reader should understand that these reasons effectively function as the basis for why different sources of protein exhibit varying degrees of anabolic potential, in regards to stimulating muscle protein accretion and promotion of exercise training adaptations over time.

## 2. Methods

This article was prepared using a narrative approach. The purpose of the review was to evaluate and review the current literature that has examined the potential impact of various plant proteins on exercise training adaptations and recovery. A range of databases, including PubMed, Medline, Google Scholar, EBSCO-host were used to search for articles for this review paper and were last accessed on 4/10/21. Inclusion criteria consisted of those studies that involved human research participants and use of at least one source of plant protein as a primary investigative agent in the study. Studies involving both acute and prolonged models were included with the majority of acute studies focusing on changes in amino acid concentration and rates of muscle protein metabolism. Prolonged studies commonly highlighted outcomes related to strength, performance, recovery, and fat-free mass. Key words routinely used to search for articles were as follows: protein, exercise, plant, oat, potato, wheat, soy, rice, pea, animal, whey, casein, beef, resistance training, strength, body composition, and MPS. Articles were chosen for inclusion based on the information they outlined and were incorporated throughout this paper. Further citations were found, evaluated, and incorporated from the bibliographies of the selected literature.

## 3. Acute Studies Using Plant Proteins and Exercise

The acute post-prandial anabolic response of an ingested protein is largely mediated by its amino acid content, with essential amino acid content, leucine, in particular, being a key driving force [8,9]. While preferences for certain protein sources may be influenced by moral beliefs, environmental considerations, dietary preferences, or assumptions regarding subsequent health outcomes, differences in amino acid content across protein sources dictate the anabolic properties of the protein. Moreover, consistent increases in MPS throughout the day have been shown to be advantageous for maximizing skeletal muscle protein accretion over time [38]. As such, it has long been suggested that higher quality sources of dietary protein confer a greater potential to increase skeletal muscle accretion compared to lower quality sources of protein.

Animal and plant-based proteins are commonly characterized by their ability to influence postprandial amino acid profiles and in their capacity to modulate rates of MPS post-ingestion. When one considers the substantial growth in popularity of plant-based diets, a number of studies have therefore examined the acute responses to a bolus ingestion of protein from varying plant-based sources [4,24,30,34,39,40,41,42,43,44], either compared to isonitrogenous animal-derived protein sources or when consumed at higher doses of total protein. Moreover, these studies often examine differences in anabolic properties both at rest or post-resistance exercise to further examine the anabolic potential or synergistic benefits when combined with exercise modalities [4,24,42]. For example, Wilkinson et al. [39] noted greater net balance in protein levels after milk ingestion compared to an isonitrogenous soy beverage, which also equated to a greater increase in fractional synthetic rate (0.10 ± 0.01 vs. 0.07 ± 0.01%/h; *p* < 0.05). Similarly, Tang et al. [24] observed a greater increase in blood EAA, branch-chained amino acid and leucine concentrations following ingestion of a whey protein hydrolysate compared to both micellar casein and soy protein isolate. Subsequently, MPS was 93% greater after consumption of whey protein compared to casein, and 18% greater than soy after exercise. These results indicated that, at rest, whey protein may elicit a more robust anabolic response immediately post-ingestion compared to casein and soy. In response to exercise, whey protein again stimulated MPS rates that were greater than both soy and casein protein, while soy was found to be greater than casein. Using a short-term supplementation protocol of 14 days, Kraemer and investigators [40] reported an attenuation of post-exercise increases in testosterone following ingestion of soy protein compared to whey protein while whey protein blunted the release of cortisol post-exercise in resistance trained males. Yang et al. [4] extended these findings and determined that a 20-g dose of soy protein isolate elicited a myofibrillar protein synthetic response that was significantly less than an equivalent dose of whey protein, but more importantly that the rates observed from a 20-g dose of soy protein were not significantly increased from consuming no protein. When the dose of protein was increased to 40 g, whey protein elicited significantly greater rates of myofibrillar protein synthesis when compared to rates observed from soy ingestion at the same dose. Finally, the 40-g dose of soy was able to demonstrate significantly greater rates of MPS when compared to when no protein was ingested. Collectively, results from these studies highlight the superiority of animal proteins (milk, whey, and casein) at stimulating acute increases in MPS rates both at rest and after exercise when compared to soy ingestion.

To accommodate the growing demand for plant-based diets, several plant protein sources have appeared in the marketplace. In this respect, acute amino acid absorption responses to a rice protein isolate identified a 6.8% lower total amino acid concentration area under the curve in rice protein isolate when compared to a whey protein isolate, but this difference was not statistically significant. Additionally, area under the curve values for essential and nonessential amino acids were not different between the two protein conditions. The time to reach peak concentration was faster with whey protein ingestion for the essential amino acids, non-essential amino acids, and total amino acids. Interestingly, however, the time to reach peak concentration for leucine was faster for rice protein isolate ingestion versus whey protein isolate ingestion [44].

In addition to this work, several studies have also assessed acute changes in MPS rates in addition to amino acid absorption to acute doses of oat, potato, peanut, and wheat protein [41,42,43,44,45,46]. For example, Lamb et al. [46] did not observe a difference in 24-h myofibrillar protein synthetic rates in subjects who received a peanut protein powder supplement versus those who received no supplement, following a bout of resistance training in older adults (59 ± 8 years). It is possible that a greater amount of peanut protein may be required in older adults to elicit meaningful post-prandial anabolic properties. In a similar manner, Gorissen et al. [43] observed greater increases in post-prandial plasma essential amino acid concentrations after whey protein ingestion (2.23 ± 0.07 mM) compared to casein (1.53 ± 0.08 mM) and wheat protein (1.50 ± 0.04 mM) (*p* < 0.01). Further, a greater increase in myofibrillar protein synthesis rate was observed after casein protein ingestion compared to whey protein (0.050% ± 0.005%/h vs. 0.032% ± 0.004%/h) (*p* = 0.003). Interestingly, post-prandial increases in plasma leucine concentrations were greater after whey protein ingestion compared to more than double the amount (60 g) of wheat protein (peak value: 580 ± 18 compared with 378 ± 10 mM, respectively; *p* < 0.01), despite comparable leucine concentrations per serving (~4 g). Another plant-based source of protein, potato protein, has a relatively high essential amino content compared to other protein sources [22], when expressed as a percent of total protein. A recent study by Oikawa et al. [42] indicated that consumption of 25 g of potato protein twice daily (1.6 g/kg/day total protein) increased myofibrillar protein synthesis at rest and in an exercised limb beyond that observed following consumption of a control diet (0.8 g/kg/day total protein) in young women (20.5 ± 3 years). Most recently, Pinckaers et al. [41] reported similar increases in post-prandial myofibrillar protein synthesis rates following consumption of a 400 mL beverage containing either 30 g of milk protein concentrate, 30 g of wheat protein hydrolysate, or 15 g of wheat protein hydrolysate plus 15 g of milk protein concentrate in young males (23 ± 3 years). Thereby indicating that wheat protein can elicit comparable anabolic properties as milk protein, when consumed in equal amounts. Collectively, these studies indicate that while several plant-based protein sources may elicit post-prandial increases in essential amino acid concentrations and subsequent increases in myofibrillar protein synthesis rates, these effects are likely to be less than or equal to what is observed following ingestion of comparable amounts of whey or casein protein. However, more research is warranted to investigate some of the newer formulations of plant-derived protein powders, such as rice, oat and potato protein and how their acute anabolic properties may influence adaptations over time; particularly when consumed in conjunction with other nutrients or exercise regimens. Overall, recent evidence indicates that when quantifying the anabolic efficiency (net protein balance/caloric intake), beef displayed greater efficiency values compared to eggs, pork loin, tofu, kidney beans, peanut butter and mixed nuts [47]. As such, animal-based sources of protein may serve as a more efficient protein source, when taking into consideration the overall energy content of a food item or meal. As a result, there has also been an interest in the development of strategies to augment the anabolic properties of plant proteins to compensate for a lower anabolic potential, a topic which will be discussed later in this review. A summary of studies to date that have examined differences in the acute anabolic response to animal and plant-based sources of protein is presented in Table 1.

## 4. Prolonged Studies Using Plant Proteins and Exercise

Up until 2013, the only research involving plant protein ingestion and regular resistance training across several weeks was completed using soy [48,49,50,51,52]. A summary table of the results of all studies which have compared a plant protein to an animal protein source (usually whey) over several weeks while completing resistance training can be found in Table 2. Brown and colleagues [48] had 27 college-aged males who were enrolled in a university weight training class consume, in a double-blind fashion, a protein bar containing either 33 g of whey or soy protein while a third group completed the training, but did not consume either bar. Over nine weeks all participants completed the resistance-training program that consisted of 3 sets of 4–6 repetitions two days per week and incorporated 14 different exercises that targeted all major muscle groups. The two protein groups gained similar amounts of lean body mass while the group that only completed the resistance training did not gain any lean mass. Candow et al. [49] used a similar study approach whereby 27 untrained healthy men and women supplemented with isocaloric doses of either whey or soy protein while following a whole-body, 4 days per week resistance training program for six weeks. Each protein source was delivered in two equal doses on training days before and after each workout while on non-training days, three equal doses were taken and spread evenly across the day. The total daily protein dose was 1.2 g/kg/day. Thus, a 70-kg individual would have consumed 84 total grams of protein per day or an estimated 28–42 g per dose. Each exercise was performed in 4–5 sets of 6–12 repetitions at an intensity of 60–90% 1 RM. Again, both sources of protein supplementation increased strength gains and accretion of lean tissue when compared to the carbohydrate control group, but no differences were identified between the plant (soy) and animal (whey) source of protein. Hartman et al. [51] had young men complete a weekly resistance training program for 12 weeks while consuming either a soy or skimmed milk beverage immediately and one hour after each workout (delivering 35 g of protein for each condition) and found that greater gains (*p* < 0.05) in fat-bone-free mass occurred with the skimmed milk group (3.9 kg, 6.2%) than what was observed in the soy group (2.8 kg, 4.4%). In 2009, Denysschen et al. [50] supplemented 28 overweight male subjects (body mass index of 25–30 kg/m^2^), all with total serum cholesterol > 200 mg/dL with either a placebo, soy, or whey protein. The whey, soy, and carbohydrate (placebo) all contained approximately 26 g and were administered in a randomized, double-blind fashion while each participant completed a 12-week supervised resistance training program. In accordance with the Brown and Candow studies, all three groups experienced significant increases in strength and fat-free mass. In addition, this study also illustrated similar decreases in percent body fat, waist-to-hip ratio, and total cholesterol in all three groups. Volek and investigators [52] randomized non-resistance trained men and women to consume either 24 g of whey protein, 24 g of soy protein, or 24 g of a carbohydrate control while completing a supervised and periodized resistance training program over a nine-month period. Lean body mass gains in the individuals consuming whey protein were found to be significantly greater (~3.3 kg) than what was observed in the soy (~1.8 kg) and carbohydrate (~2.3 kg) groups.

In 2013, Joy and colleagues [53] were the first to examine the impact of rice protein for its ability to impact resistance training adaptations and this was also one of the first times a plant protein source other than soy was assessed for its potential to impact resistance training adaptations. This study randomized 24 healthy males in a double-blind fashion to ingest either 48 g of whey protein or rice protein isolate. The participants supplemented for eight weeks and followed a three day per week resistance training program. Significant increases in fat-free mass, maximal strength, and lower-body power occurred in both protein groups, but no differences in changes were observed between the two protein sources. The protein dose in this study (48 g) was chosen to ensure that adequate amounts of leucine were being delivered in both the rice and whey protein groups. Results of the study revealed similar outcomes as seen previously with soy, whereby similar short-term changes in resistance training adaptations were observed between plant and animal protein sources. As a follow-up, Moon and colleagues [54] had 24 healthy, resistance-trained males perform a four days per week split-body, linearly periodized resistance training program (3–4 sets of 6–10 RM loads) for ten weeks. In a randomized, double-blind fashion, participants began supplementing daily after completing two weeks of resistance training with 24-g doses of either a rice or whey protein concentrate. The chosen dose in this study was intended to deliver a dose of rice protein that just met what has been considered by many to be the minimum amount of leucine (~2.0 g) to stimulate protein translation [5,55]. As seen previously, significant increases in body mass, total body water, lean mass, fat-free mass, maximal upper body strength, upper body volume, and maximal lower-body strength were observed throughout the study in both groups. No differences between the two protein groups were observed for any of these outcomes, leading the authors to conclude that the observed resistance training outcomes were similar between the two protein conditions. These results are significant in the sense that this was one of the first studies to illustrate similar potential of a plant protein source to elicit changes in strength and body composition, using a smaller dose of a plant-based protein over a short period of exercise training and supplementation. Moreover, the findings also support the notion that as long as an efficacious dose of leucine and essential amino acids are ingested, that favorable exercise training adaptations can result from a plant protein source.

In 2015, Babault and colleagues [56] investigated the impact of a pea protein on changes in exercise training adaptations. Over 12 weeks, 161 males between the ages of 18–35 years completed upper body resistance training while supplementing with either pea protein, whey protein, or placebo. The total protein dose was 50 g per day that was divided up into two 25-g doses each day. Increases in muscle thickness tended (*p* = 0.09) to be greater in the pea protein group when compared to changes observed in the whey and placebo groups. Interestingly, when a sub-analysis was completed of those participants who had the lowest strength levels to start the study, pea protein supplementation exhibited a greater ability to increase muscle thickness levels. These results led the authors to conclude that a pea protein supplement could serve as an alternative to whey protein. Reidy and investigators [57] were the first to investigate the ability of a blend of soy and dairy proteins to increase strength and body composition. In randomized, double blind fashion, 58 participants consumed a 22-g dose of a soy-dairy protein balance, 22 g of whey protein isolate or an isocaloric carbohydrate placebo. Participants supplemented for 12 weeks while completing a resistance-training program. All groups experienced increases in lean mass, with the changes observed in the soy-dairy protein tending to be greater than what was seen in carbohydrate (*p* = 0.09), with no differences being observed between the whey protein isolate (*p* = 0.55). Changes in strength were similar between all groups. Muscle thickness was significantly increased in all participants with a trend being observed for differences between groups (Mean: 0.92 kg, 95% CI: −0.12, 1.95 kg, *p* = 0.09). In 2017, Mobley et al. [58] reported no differences between groups for the observed changes in strength, body composition or various tissue attributes of skeletal muscle or adipose tissue after supplementing and resistance training for 12 weeks. In this study, 75 untrained college-aged males were randomly assigned to consume a carbohydrate placebo, whey protein hydrolysate, whey protein concentrate, or a soy protein concentrate. A similar outcome was reported for Lynch and colleagues [59] who randomly supplemented 48 untrained men and women for 12 weeks with either 19 g of whey protein isolate or 26 g of soy protein isolate; protein dose amounts that both delivered 2 g of leucine. In both protein groups, body mass, lean mass, peak extension and flexion torques all increased significantly in both groups while muscle thickness tended to increase after 12 weeks of resistance. As seen in previous studies, no differences between the two protein groups were observed for the measured outcomes.

Hevia-Larrain and colleagues [60] have been one of the only research groups to examine the impact of habitually consuming a plant-based versus an omnivorous diet. This project examined the impact of protein-matched diets on resistance training adaptations in 38 young men who were physically active, but naïve to resistance training. Habitual (longer than 12 months) vegans or omnivores were assigned to a protein group and were given supplemental protein (in the form of soy protein for vegans and whey protein for omnivores) to achieve a daily protein intake of 1.6 g/kg/day. For 12 weeks, each participant resistance trained their lower-body musculature two times per week and has strength, muscle mass and cross-sectional area assessed. All measured outcomes improved in both groups across the 12-week study protocol, but there were no differences between the two protein groups. These outcomes support previous work that indicates that plant proteins, when provided as part of daily protein intake that meets daily needs, can lead to comparable improvements in strength and body composition outcomes when compared to animal proteins.

In summary, a growing number of studies have evaluated the ability of plant protein sources to stimulate resistance-training adaptations in comparison to the adaptations seen with an animal source of protein. When viewed collectively, the majority of published studies, as designed, consistently indicate that plant proteins can deliver similar changes in strength and body composition when strategies are taken to either equate the amount of leucine being delivered or ensuring that enough leucine and the other essential amino acids are being delivered. The majority of studies completed thus far have been 8–12 weeks in duration and this may function as an important consideration when interpreting this literature. A key exception to this was seen with Volek et al. [52] who reported more favorable adaptations after whey protein ingestion when compared to an identical dose of soy protein after 9 months of training. Thus, it remains quite possible that while studies performed of shorter durations are reporting equivalence within these established delimitations that if future studies are performed for longer time periods (4–6 months or longer) that different outcomes may result. To support this notion, the Moon et al. [54] study reported no differences in strength and body composition changes after eight weeks of supplementing with a 24-g dose of either rice or whey protein with a total daily protein intake of 1.4–2.0 g/kg/day, however, the largest mean changes from baseline were observed in the whey protein group.

## 5. Recovery Considerations for Plant Protein Sources

Additional research has examined the ability of various plant-based proteins for their ability to influence post-exercise protein kinetics and recovery [62,63,64,65]. For example, Kritikos et al. [62] recently examined differences in recovery kinetics following speed endurance training in male soccer players after ingesting whey or soy protein. The authors concluded that both whey and soy protein were able to mitigate reductions in field-based performance during successive speed-endurance training sessions, with neither protein source appearing to have an effect on exercise-induced muscle damage or markers of oxidative stress. Using an eccentric muscle damage model, Nieman and investigators [64] compared the ability of whey or pea protein to mitigate decrements in force production and increases in markers of swelling, muscle damage, and inflammation. A 90-min bout of eccentric exercise in 92 untrained, non-obese males was used to invoke muscle damage. The participants were divided into three groups: placebo (water), whey protein (0.9 g/kg divided into three doses per day), and pea protein (0.9 g protein/kg divided into three doses per day) and changes in force production, power, and blood markers were assessed each day for five consecutive days. Following muscle damage, Whey protein significantly attenuated increases in blood-based markers of muscle damage while the changes observed in pea protein were not significantly different than what was observed in the water condition. No differences, however, were identified between the magnitudes of differences observed in the two protein groups. Xia et al. [63] examined the effects of oat protein supplementation on markers of muscle damage and inflammation in addition to measures of performance following downhill running. After 14 days of supplementation with 25 g per day of oat protein, an attenuation of the observed increases in eccentric exercise-induced muscle soreness and serum concentrations of IL-6, creatine kinase, myoglobin, and C-reactive protein were observed. A marked reduction in lower limb edema, in addition to a lesser reduction in muscle strength, knee-joint range of motion and vertical jump performance was observed following oat protein supplementation when compared to placebo.

In contrast with the previous findings that suggested a favorable ability of protein to promote recovery, Saracino and researchers [65] had 27 recreationally active, middle-aged men complete 5 sets of 15 repetitions using eccentric contractions the knee extensors and flexors. Starting the same day as which muscle damage occurred, participants ingested equivalent doses (40 g) of whey protein hydrolysate, whey isolate, or a rice and pea protein combination in addition to a placebo group 30 min prior to going to sleep and did this supplementation regimen again for the next two nights. Nutrient intake was standardized to ensure adequate daily protein and a series of circumference, soreness, muscle damage markers and strength measures were taken for 72 h after completion of the exercise bout. While widespread and predictable changes in the measured outcomes occurred in response to the exercise bout, no differences were identified between any of the supplementation groups. As such, the authors concluded that pre-sleep supplementation protein ingestion, regardless of protein source, did not aid in muscle recovery from muscle-damaging exercise. The results from the Saracino study align with previous indications by Pasiakos et al. [66], who concluded in their meta-analysis that added protein may exert limited benefit in terms of promoting recovery and reducing muscle damage and soreness. In this respect, it is difficult to draw conclusions across studies that investigated the effects of only plant or animal-based proteins in isolation, rather than comparing multiple protein sources within the same study. As such, contextual factors such as exercise modalities, differences in protein metabolism assessment techniques and subject characteristics may confound any further ability to draw conclusions across the literature regarding a superior effect of one protein sources over the other. Consequently, more studies are needed that examine the potential of single or blended sources of plant protein in comparison to animal sources for their ability to differentially impact performance or various recovery metrics in response to challenging doses of exercise. A summary table of all studies which have compared some aspect of exercise recovery between a plant and animal source of protein can be found in Table 3.

## 6. Considerations for Older Adults

It is well-established that as individuals age their rate of muscle mass loss (i.e., sarcopenia) [67,68] and muscle strength and function loss (e.g., dynapenia) [69] both increase. Accepted countermeasures for these changes are an increase in weight-bearing (resistance) exercise and an adequate delivery of protein and amino acids. In this respect, several studies are now available that have examined the impact of protein ingestion in older populations. For example, post-prandial MPS rates after ingesting 24 g of soy protein have been shown to be lower in older adults when compared to beef protein ingestion [70]. Moreover, Yang and colleagues [4] examined the dose-response impact of soy protein ingestion in older adults and found that doses of up to 40 g of soy protein failed to elevate MPS rates from basal (fasting) levels. In consideration of soy ingestion, these results are important as they seemingly suggest that even a large dose (40 g) may fail to appropriately stimulate MPS rates. Other studies have examined the impact of plant-based foods in elderly women [71] and concluded that net protein synthesis was lower during a high vegetable protein diet versus a high animal protein diet. Moreover, Gorissen et al. [43] had 60 healthy older men consume one of four sources of protein in a 35-g dose: whey, micellar casein, wheat, or wheat protein hydrolysate or a 60-g dose of wheat protein hydrolysate (an amount that deliver equivalent amounts of leucine as in the 35 g dose of whey). Postprandial increases in plasma leucine were highest after ingesting whey while myofibrillar protein synthesis increases were greater in whey and casein while the 60-g dose of wheat matched rates of myofibrillar protein synthesis. When viewed in concert with the findings of Yang et al. [4], these outcomes highlight the need for older individuals to either consume larger doses of plant proteins or for strategies to be implemented that increase the anabolic potential of the plant protein dose. Practically speaking, these results are troubling and seemingly work against the age-related loss of appetite and enjoyment from food that occurs with advancing age [72].

Finally, two studies have examined the impact of combining different sources of plant proteins in combination with resistance training in older adults to identify the impact that plant protein consumption may have on changes in strength and body composition. Briefly, Thomson et al. [61] compared changes in strength and body composition in both soy protein and dairy protein (both consumed in dosages of 27 g/day and a total protein intake of 1.2 g/kg/day) in a group of older (61.5 ± 7.4 years) adults. After 12 weeks, both groups experienced increases in strength and fat-free mass, but no differences between the two protein sources were found. Similarly, Lamb and colleagues [46] randomized 39 older (58 ± 8 years), untrained men and women to consumed either a defatted peanut protein powder (30 g protein, 9 g essential amino acids) or no supplement at all. Hypertrophy and performance were assessed six and ten weeks after supplementation and no changes in fat mass, lean, or percent body fat were found between the groups. An increase in vastus lateralis thickness was observed in the peanut protein group when compared to the no-supplement controls and peak power increased in the peanut powder group. The authors concluded that a defatted peanut protein powder may positively impact resistance training adaptations seen in a group of healthy, older previously untrained men and women. More research is needed to help identify what differential impact, if any, plant protein sources may hold over animal sources of protein.

## 7. Increasing the Anabolic Potential of Plant Sources

Several strategies exist to increase the anabolic potential of various protein sources. These strategies include but are not exclusive to increasing daily protein intake, co-ingestion of plant proteins with amino acids or other nutrients, supplementing plant sources with those amino acids deemed to be low or limiting, and blending various protein sources together. Certainly, the easiest solution to overcome the lower levels of amino acids and digestibility is to increase the size of protein dose. In this respect, studies in younger subjects [15,73] illustrate that a dose of 20–25 g of protein (0.25 g/kg body/dose) can maximize MPS using animal sources. When using plant protein sources, as highlighted by other studies [4,43], larger doses are likely needed to maximize the MPS response. While accepted to be a simple recommendation, pragmatic aspects must be considered as sometimes larger doses might be challenging for people to consume due to larger volume of fluid, higher fiber intakes (common in plant-based foods), or food being needed to ingest, particularly for older individuals.

Another strategy that needs further exploration involves the co-ingestion of plant proteins with various nutrients to help increase the anabolic potential of plant protein, particularly in those populations that need more protein and/or may not be consuming enough protein. Towards this end, previous research has indicated that consuming omega-3 fatty acids with an amino acid infusion surrounding resistance exercise can heighten anabolic sensitivity of skeletal muscle and increase rates of MPS [74,75]. This practice, however, has yet to be evaluated in an exercise training model in combination with plant protein consumption. Nonetheless, these results are of great interest and future research should seek to explore this approach with plant sources of protein to determine if the increased anabolic sensitivity also occurs with intact plant ingestion and then if this translates to greater gains in health and resistance training adaptations.

As highlighted earlier, the leucine content of feeding has been shown to be of critical importance in terms of stimulating MPS [14,15]. In this respect and on a per gram basis, plant sources have lower amounts of leucine as well as many of the essential amino acids [22]. To overcome these shortcomings, researchers have explored the impact of consuming smaller doses of protein but fortifying the dose with added leucine or other limiting amino acids. For example, Churchward-Venne and colleagues [76] added leucine to a small dose (6.25 g) of whey protein to match the leucine that was delivered in a 25-g dose of whey protein. They demonstrated this approach was effective at stimulating fed rates of MPS, but the 25-g dose of whey protein better sustained exercise-induced rates of MPS. While the approach has yet to be examined using a plant protein sources, previous studies [77,78] that combined plant proteins with leucine or all three branched-chain amino acids have illustrated favorable changes in MPS and how certain amino acids are metabolized inside various tissues. Future work should build upon these approaches to examine their efficacy at promoting favorable adaptations to exercise training.

A commonly proposed solution to overcoming the shortcomings associated with plant protein intake center upon mixing the plant source with an animal source or another plant source [79]. Using this approach, acute MPS responses were assessed after ingesting a protein blend of 25% whey protein, 25% soy protein, and 50% casein protein and completion of a single bout of lower-body resistance exercise. When compared to an isonitrogenous dose of whey protein in young, healthy males, the protein blend increased mixed MPS rates to a similar magnitude as what was observed with whey protein consumption [80]. This acute study was followed up using a 12-week resistance-training model whereby Reidy and colleagues [57] supplemented 68 young, healthy males daily with 22-g doses of either a blend of soy and dairy proteins, an isocaloric carbohydrate control, or a protein-equated whey protein group while performing a supervised resistance training program three days per week. When compared to a carbohydrate control, the protein blend tended to increase lean mass while no change was observed in the whey protein group. This led the authors to conclude that consumption of a protein blend slightly enhanced gains on whole-body as well as arm lean mass while strength changes were not different between groups. For many people, however, a protein blend consisting of only 25% soy protein and 75% animal protein will not be acceptable. Thus, depending on the underlying reason for exclusively selecting plant-based sources of protein, it may not be practical for individuals to combine plant- and animal-based proteins. In this respect, blending multiple plant protein sources has been explored to maximize amino acid delivery while also creating a blend that is 100% plant-derived. Currently, no data exists using this approach to identify acute changes in muscle protein synthetic responses or changes in resistance training adaptations after several weeks of administration. More research in this area should be considered.

Another strategy to heighten the potential impact of plant protein ingestion could center upon the timing or proximity of when nutrients are consumed relative to the exercise. The concept of nutrient timing is not new and current position stands on the topic have thoroughly discussed the literature surrounding its efficacy [81]. As highlighted previously, resistance-based exercise induces a period of sensitization in skeletal muscle that enhances the anabolic properties of protein ingestion [82]. As a result, more of the amino acids consumed from dietary sources are directed towards incorporation into peripheral tissues versus splanchnic extraction, which facilitates greater increases in MPS rates [83]. This heightened sensitivity has been shown to persist for up to 24 h after completion of an exercise bout [82]. Consequently, when plant protein feedings are provided, which depending on many factors discussed throughout this paper may result in a smaller bolus of amino acids being delivered, they may still be able to instigate meaningful increases in MPS rates if they are ingested during this period of heightened sensitivity. Currently, no research has explored the potential for timing with ingestion of plant protein sources and future studies should seek to determine the extent to which (if any) these strategies can help improve adaptations commonly seen from resistance exercise. Finally, recent studies by Stecker [84] and Jäger [85] have provided evidence that adding various strains of a probiotic to an animal source of protein and a plant source of protein, respectively, may favorably impact the appearance of various amino acids into the bloodstream when coingested with protein.

## 8. Conclusions and Future Directions

The popularity of plant proteins has grown substantially in recent years. Initial research that examined the acute impact of various sources of protein at stimulating MPS clearly points towards an advantage for the highest quality protein sources, which are viewed to be those that are derived from animal sources. As such, animal proteins were strongly advocated for health and performance outcomes while plant sources of protein were viewed to be inferior at helping exercising individuals achieve their exercise training goals. Only recently have studies begun to appear that have compared the ability of various animal and plant protein sources regarding facilitating increases in strength, endurance, power, fat-free mass accretion, and recovery over the course of several weeks of exercise training and supplementation. From this prolonged data, a consistent pattern has appeared which suggests that when total daily protein intake is achieved at levels recommended for exercising athletes (1.4–2.0 g/kg/day) [5,7,11], the source of protein does not function as a determining factor in the observed outcomes.

Two key considerations stemming from this conclusion, however, must be considered. First, only one study to date [60] has made such comparisons in study participants who were habitually consuming either plant or animal sources of protein. This point is not made to detract from the significance of the other published findings, but the majority of studies that have provided a daily dose of a plant protein have done so with individuals consuming diets mixed with various animal protein sources. Thus, for a 180-pound (81.7 kg) individual who is consuming 1.5 g/kg/day of protein, a daily 25-g dose of plant protein represents approximately 20% of that individual’s daily protein intake and one can reasonably question how much impact changing the source of just 20% of the daily protein delivered will impact overall outcomes. Second, nearly all studies (acute and prolonged) have utilized free amino acid mixtures or isolated protein powders while the majority of nearly all dietary protein is consumed as some form of mixture of macro- and micronutrients. More research needs to continue to explore how the matrix of nutrients found in single foods and meals impacts these outcomes. The future is bright, however, for plant proteins, as strategies have been articulated in this paper and others [16,23] regarding various strategies that can be considered to help increase the quality of each plant protein feeding. In this respect, more research is needed to identify if co-ingestion of plant proteins with various nutrients can heighten desired physiological adaptations by exercising individuals. Furthermore, research should explore how changes in plant protein manufacturing (hydrolyzing, heat treatment, etc.) as well as the timing or pattern of how the protein is administered, particularly in reference to completion of resistance exercise, may confer certain advantages.

## Figures and Tables

**Table 1 nutrients-13-01962-t001:** Summary Table of Acute Responses to Plant Protein Ingestion.

Reference	Participants	Design	Study Duration	Dosing Protocol	Exercise Program	Primary Variables	Key Findings
Wilkinson et al. 2007 [39]	8 healthy males (21.6 ± 0.3 years.)	RCT, crossover (2 groups) Milk (*n* = 8) Soy (*n* = 8)	1 trial visit per condition 7-day washout	Macronutrient-matched soy or milk beverages (18 g protein)	Lower body exercise bout	Protein kinetics Net muscle protein balance	↓ Net balance (AUC) after soy ingestion vs. milk ↓ Fractional synthesis rate in muscle after soy consumption vs. milk
Tang et al. 2009 [24]	6 healthy young men (22.8 ± 3.9 years.)	RCT, crossover (3 groups) Whey (*n* = 6) Casein (*n* = 6) Soy (*n* = 6)		10 g of EAA in the form of: Whey, casein and soy protein	Unilateral lower-body exercise	Mixed muscle protein fractional synthetic rate (FSR) Blood EAA	↓ Blood EAA, BCAA, and leucine concentrations following soy ingestion compared to whey ↓ MPS (~18%) after soy consumption vs. whey after exercise ↑ MPS (~64%) with soy consumption at rest and following resistance exercise (69%) vs. casein
Yang et al. 2012 [4]	30 elderly men (71 ± 5 years.)	RCT (3 groups) Control Soy 20 g Soy 40 g	1 trial visit per group 4 h post-protein consumption	20 g or 40 g of soy protein isolate Compared to previous responses from similarly aged men who had ingested 20 g and 40 g of whey protein isolate	Acute bout of unilateral knee-extensor resistance exercise prior to ingesting no protein	Myofibrillar protein synthesis (MPS)	↑ Whole-body leucine oxidation for S20 vs. W20 ↔ in both exercised and non-exercised leg muscles for S20 vs. 0 g ↓ MPS post S40 under both rested and post-exercise conditions vs. W40 ↑ MPS post S40 than 0 g under post-exercise conditions
Kraemer et al. 2013 [40]	10 resistance trained males (21.7 ± 2.8 years.)	RCT, crossover (3 groups) Whey protein isolate Soy protein isolate Maltodextrin	14 days	20 g	Acute heavy resistance exercise test consisting of 6 sets of 10 repetitions in the squat exercise at 80% of the subject’s 1 RM	Sex hormones post resistance training	↓ Testosterone responses following supplementation with soy protein ↔ SHBG concentrations between experimental treatments ↔ in estradiol concentrations between groups
Purpura et al. 2014 [44]	10 trained male subjects (22.2 ± 4.2 years.)	RCT, crossover (2 groups) Rice protein Whey protein	2 trial visits per condition (7-day washout)	48 g isonitrogenous and isocaloric	N/A	Plasma concentrations of amino acids	↑ Tmax for RPI for EAA, non-EAA, and total amino acids ↔ For AUC between conditions ↔ for Cmax between conditions ↑ Cmax faster for leucine in the RPI group.
Gorissen et al. 2016 [43]	60 healthy older men (71 ± 1 years.)	RCT (5 groups) Wheat (*n* = 12) WPH35 g (*n* = 12) Casein (*n* = 12) Whey (*n* = 12) WPH60 g (*n* = 12)	1 trial visit per group. 240 min	35 g or 60 g	N/A	Postprandial increase in plasma EAA concentrations	↓ Postprandial increase in plasma EAA concentration after ingesting WPH-35 vs. Whey-35 ↓ Myofibrillar protein synthesis rates after ingesting WPH-35 vs. MCas-35 ↓ Postprandial increase in plasma leucine concentrations after ingesting WPH-60 vs. Whey-35
Oikawa et al. 2020 [42]	24 healthy young women (21 ± 3 years.)	RCT, single blind (2 groups) PP (*n* = 12) Control (*n* = 12)		25 g of potato protein (PP) twice daily (1.6 g/kg/d total protein) (CON) (0.8 g/kg/d total protein) for 2 weeks.	Unilateral RE (~30% of maximal strength to failure) was performed thrice weekly with the opposite limb serving as a non-exercised control (Rest)	Myofibrillar protein synthesis	↑ MPS at Rest, and in the Exercise limb following PP ingestion ↑ MPS in CON vs. baseline after Exercise only.
Pinckaers et al. 2021 [41]	36 males (23 ± 3 years.)	RCT, parallel-group design 3 groups (*n* = 12/group)		30 g milk protein (MILK) 30 g wheat protein (WHEAT) 30 g blend combining 15 g wheat plus 15 g milk protein (WHEAT+MILK).	N/A	Post-prandial plasma amino acid profiles Myofibrillar protein synthesis rates	↓ Post-prandial plasma EAA concentration post WHEAT vs. MILK ↔ Post-prandial plasma EAA concentration post MILK and WHEAT+MILK ↔ Post-prandial myofibrillar protein synthesis rates between MILK vs WHEAT ↔ Post-prandial myofibrillar protein synthesis rates between MILK vs WHEAT+MILK

↔ = No difference (*p* > 0.05) change; ↑ = Greater increase (*p* < 0.05) over control or other condition/intervention. ↓ = Lesser or decrease (*p* < 0.05) over control or other condition/intervention. AUC = area under the curve; MILK = Milk protein; MCas = Micellar casein; WPH = wheat protein hydrolysate; RPI = Rice protein isolate; WPI = Whey protein isolate; EAA = Essential amino acid; NEAA = non-essential amino acid; TAA = total amino acid; AUC = Area under the curve; C_max_ = maximum concentration; t_max_ = time at which maximum concentration was reached. Nmol/mL = nanomole/milliliter; PP = Potato protein. 1 RM = one repetition maximum. N/A = Not applicable as no exercise protocol was used.

**Table 2 nutrients-13-01962-t002:** Summary Table of Prolonged (Training) Examining Exercise Training Adaptations Using Plant Protein Sources.

Reference	Participants (Age)	Design	Study Duration	Dosing Protocol (Timing)	Exercise Program	Primary Variables	Key Findings
Babault et al. [56]	161 males (18–25 years)	RCT (3 groups) Control (*n* = 54) Whey (*n* = 53) Pea (*n* = 53)	12 weeks	50 g pea/day (two 25 g doses)	RT 3×/week	Muscle thickness Strength	↑ Bicep thickness ↑ 1-RM Strength
Brown et al. [48]	27 healthy, college-aged males (19–25 years)	RCT (3 groups) Control (*n* = 9) Whey (*n* = 9) Soy (*n* = 9)	9 weeks	33 g soy/day (11 g dose 3x/d)	RT 2×/week	Body comp	↑ Fat-free mass ↓ Percent body fat
Candow et al. [49]	27 non-active males and females (18–35 years)	RCT (3 groups) Control (*n* = 9) Whey (*n* = 9) Soy (*n* = 9)	6 weeks	1.2 g soy/day (3 daily doses)	RT 4×/week	Body comp Strength	↑ Fat-free mass ↑ Strength
DeNysschen et al. [50]	28 overweight males (21–50 years)	RCT (3 groups) Control (*n* = 9) Whey (*n* = 10) Soy (*n* = 9)	12 weeks	26 g soy/day(Post-workout)	RT 3×/week	Body comp Strength Anthropometrics	↑ Fat free mass ↓ Percent body fat ↑ Strength ↓ Waist-to-hip ratio
Hartman et al. [51]	57 healthy males(18–30 years)	RCT (3 groups) Control (*n* = 19) Milk (*n* = 18) Soy (*n* = 19)	12 weeks	17.5 g soy/day (Post-workout)	RT 5×/week	Body comp Strength Muscle fiber size	↑ Fat-free mass ↔ Strength ↑Muscle fiber area
Hevia-Larrain et al. [60]	38 untrained young males (18–35)	RCT (2 groups) Vegans (*n* = 19) Ominivores (*n* = 19)	12 weeks	1.6 g/kg/day (Soy or Whey)	RT 2×/week	Leg muscle mass Muscle mass Muscle fiber size Strength	↑ Leg muscle mass ↑ Lean body mass ↑ VL CSA ↑ Leg press 1-RM
Joy et al. [53]	24 healthy males (18–30)	RCT (2 groups) Rice (*n* = 12) Whey (*n* = 12)	8 weeks	48 g rice/day (Post-workout)	RT 3×/week	Body comp Strength Power	↑ Fat-free mass ↑ Strength ↑ Wingate power
Lamb et al. 2020 [46]	39 non-active older males and females (50–80 years)	RCT (2 groups) Control (*n* = 19) Peanut protein (*n* = 20)	10 weeks	30 g peanut/day (1x/d)	RT 2×/week	Body comp Muscle thickness Knee flexion torque	↔ Body comp ↑ VL thickness ↑ Knee flexion torque
Lynch et al. [59]	48 non-active males and females (18–35 years)	RCT (2 groups Whey (*n* = 26) Soy (*n* = 22)	12 weeks	19 g whey or 26 g soy/day (post-workout)	RT 3×/week	Body mass Body comp Muscle thickness Knee flexion and extension torque	↑ Body mass ↑ Fat-free mass ↔ VL thickness ↑ Peak torque
Mobley et al. [58]	75 healthy, untrained males (19–23 years)	RCT (5 groups) Control (*n* = 15) Leucine (*n* = 14) WPC (*n* = 17) WPH (*n* = 14) Soy (*n* = 15)	12 weeks	39.2 g soy/day (post-workout and pre-sleep)	RT 3×/week	Strength Body mass Body comp Muscle fiber CSA	↔ Strength ↔ Body mass ↑ Muscle Mass ↑ Type I/II CSA
Moon et al. [54]	24 healthy, trained males (18–50 years)	RCT (2 groups) Whey (*n* = 12) Rice (*n* = 12)	8 weeks	24 g rice or whey/day (post-workout)	RT 4×/week	Body comp Muscular strength Muscular Endurance Anaerobic Capacity	↑ Body comp ↑ 1-RM strength ↑ Rep to fatigue ↑ Wingate power
Reidy et al. [57]	67 healthy males (18–35 years)	RCT (3 groups) Control (*n* = 23) Whey (*n* = 22) Soy (*n* = 23)	12 weeks	22 g soy or whey/day (post-workout)	RT 3×/week	Body comp Strength mCSA Muscle thickness	↑ Lean body mass ↔ 1RM strength ↔ mCSA ↔ Muscle thickness
Thomson et al. [61]	83 older adults (50–79 years)	RCT (3 groups) Control (*n* = 23) MILK (*n* = 34) Soy (*n* = 26)	12 weeks	27 g soy/day (post-workout)	RT 3x/week	Strength Body comp Physical function	↔ Strength ↑ Lean mass ↑ Physical function
Volek et al. [52]	63 untrained males and females (18–35 years)	RCT (3 groups) Control (*n* = 22) Whey (*n* = 19) Soy (*n* = 22)	9 months	24 g soy protein (Post-workout)	RT 3 ×/week	Body comp	↑ Lean body mass ↔ Fat mass

↔ = No difference (*p* > 0.05) change; ↑ = Greater increase (*p* < 0.05) over control or other condition/intervention. ↓ = Lesser or decrease (*p* < 0.05) over control or other condition/intervention. WPC = whey protein concentrate; WPH = whey protein hydrolysate; MILK = milk protein; mCSA = muscle cross-sectional area; 1 RM = one repetition maximum.

**Table 3 nutrients-13-01962-t003:** Summary Table of Studies Examining Exercise Recovery Outcomes Using Plant Protein Sources.

Author (Year)	Participants (Age)	Design	Study Duration	Dosing Protocol (Timing)	Exercise Program	Primary Variables	Key Findings
Nieman et al. [64]	92 healthy, untrained males(18–55 years)	RCT (3 groups) Control (*n* = 30) Whey (*n* = 31) Pea (*n* = 31)	5 days	0.3 g/kg/d pea or whey/day (Pre-workout)	90 min eccentric exercise bout	Strength Vertical jump Anaerobic power Muscle soreness	↔ 1 RM strength ↔ Vertical jump ↔ Wingate power ↑ Muscle soreness
Saracino et al. [65]	27 active, middle-aged males(40–64 years)	RCT (4 groups) Control (*n* = 6) WPH (*n* = 9) WPI (*n* = 6) Rice/pea (*n* = 6)	3 days	40 g rice/Pea blend/day (pre-sleep)	Lower body muscle-sdamaging exercise	MVC Muscle soreness Thigh circumference	↓ MVC ↔ Muscle soreness ↔ Thigh circumference
Kritikos et al. [62]	10 well-trained soccer players (*n* = 10)	RCT, crossover	3 days	1.5 g/kg/day whey or soy	Field-based speed training sessions	Performance Isokinetic strength MVC Lower body power Muscle damage Creatine kinase Muscle soreness	↓ Isokinetic leg strength ↓ MVC ↓ Speed ↓CMJ ↑ CK ↑ DOMS
Xia et al. [63]	16 healthy, non-active males (19.7 ± 1.1 years)	RCT (2 groups) Control (*n* = 8) Oat (*n* = 8)	19 days	25 g oat/day (post-workout)	Downhill running	Muscle soreness IL-6 Creatine kinase Leg strength Vertical jump	↓ Muscle soreness ↓ IL-6 ↓ CK ↑ 1 RM strength ↑ Vertical jump

↔ = No difference (*p* > 0.05) change; ↑ = Greater increase (*p* < 0.05) over control or other condition/intervention. ↓ = Lesser or decrease (*p* < 0.05) over control or other condition/intervention. WPC = whey protein concentrate; WPH = whey protein hydrolysate; WPI = whey protein isolate; MILK = milk protein; DOMS = delayed onset muscle soreness; CK = creatine kinase; IL-6 = interleukin-6; MVC = maximal voluntary contraction; 1 RM = one repetition maximum.

## Data Availability

Not applicable.

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
