# Peer review of "Plant Proteins and Exercise: What Role Can Plant Proteins Have in Promoting Adaptations to Exercise?"

_nutrients, 2021, doi:10.3390/nu13061962_

Round 1
Reviewer 1 Report
The review by Kersick et al. focus on the implications of plant protein consumption and plant-based diets on outcomes linked to exercise performance, associated exercise training adaptations and recovery.
The paper is in theory interesting, but is very long and difficult to read since it is a mixture of a specific literature search and general information and considerations.
Sometimes a wordy description of studies was offered. Most information are already available or may be added in Tables. there are several repetitions in the text.
Overall, the paper should be shortened by 50%
SPECIFIC COMMENTS
1.1 MUSCLE PROTEIN METABOLISM
Place the aim of the study at the endo of this paragraph.
1.2 IMPORTANCE OF ADDED PROTEIN TO THE DIET
Line 53. A more appropriate title may be “The importance of added protein to optimize training adaptation”.
Why is there so much interest in using plant proteins to promote muscle mass and function?
1.3 THE CASE FOR PLANT PROTEINS
This paragraph is useless and a few concepts of interest may be easily included in para 1.4.
1.4 QUALITY CONSIDERATIONS
I think that this paragraph needs to be made clearer and much more concise.
The difference in protein quality between and within animal and plant proteins needs to be more adequately explained in terms of digestibility and amino acid composition.
It should be underlined that the authors focused on the evaluation of single protein sources (not of mixed diets).
PDCAAS and DIAAS take into consideration both digestibility and IAA composition. If I remember correctly, PDCAAS ranges between 1 and 0. DIAAS has been specifically proposed for evaluating the protein quality of specific food souces, also in order to define claims. Digestibility is much higher in plant protein isolates. Amino acid composition of soy protein should be reported.
What are the plant proteins that have been used to optimize training adaptation? What are the plant proteins the authors are interested in? What is their protein quality? Comments on other plant proteins (legumes, cereals, etc.) could be misleading for the reader.
Leucine content should be considered and treated in this paragraph.
Line 128-129 The statement is not completely true. A score below 1 just indicates that the amount of IAA/g protein provided by a given protein is lower than the maximum one. In several plant proteins, this is mainly due to a reduced digestibility.
Line 135 Notably, the added values of DIAAS identifies which amino acids are limiting. Quite vague…
- METHODS
This section should be completely rewritten according to standard section structure: PRISMA indications, selection of papers, researchers involved, quality of papers, etc.
- ACUTE STUDIES
More than 1800 words to appraise eight papers. Too much.
Key messages are getting lost in too much text. Sometines, there is a wordy dscription of studies in a mechanistic sequence. The description of each paper needs to be shortened, avoiding the repetition of data provided in the Tables.
It is not clear to the reader whether all the studies reported in Table 1 were actually focused on physical training.
Be more concise and focus on studies related to physical exercise.
Line 229. What does “net balance in protein levels” mean?
- PROLONGED STUDIES USING PLANT PROTEINS AND EXERCISE
Be more concise and make concepts more accessible and concrete.
Key messages are getting lost in too much text. Sometines, there is a wordy dscription of studies in a mechanistic sequence. The description of each paper needs to be shortened, avoiding the repetition of data provided in the Tables.
- RECOVERY CONSIDERATIONS FOR PLANT PROTEIN SOURCES
Be more concise and make concepts more accessible and concrete.
- CONSIDERATIONS FOR OLDER ADULTS
This is another story…
The paragraph could be deleted.
- INCREASING THE ANABOLIC POTENTIAL OF PLAT SOURCES
Once more, the content of the paragraph is interesting, but the text is not straightforward nor easily readable. Focus on exercise with brief general considerations.
- Where is this paragraph?
- CONCLUSIONS AND FUTURE DIRECTIONS
To be reconsidered based on previous paragrphs and made more persuasive.
OTHERS
Use MPS for muscle protein synthesis, PDCAAS instead of PDCAAs and DIAAS instead of DIAAs.
There are some mispellings.
Author Response
REVIEWER 1
Manuscript ID nutrients-1215584
Plant Proteins and Exercise: What Role Can Plant Proteins Have in Promoting Adaptations to Exercise?
Comments and Suggestions for Authors
The review by Kersick et al. focus on the implications of plant protein consumption and plant-based diets on outcomes linked to exercise performance, associated exercise training adaptations and recovery. The paper is in theory interesting, but is very long and difficult to read since it is a mixture of a specific literature search and general information and considerations.
Sometimes a wordy description of studies was offered. Most information are already available or may be added in Tables. there are several repetitions in the text. Overall, the paper should be shortened by 50%
AUTHOR RESPONSE: We appreciate your response and time reviewing this paper. It was our intention however to provide more context and nuance as we feel this details helps people to understand foundational tenets of the topic and for future research to be better planned. Regardless, we understand your feelings and have worked to shorten the paper.
SPECIFIC COMMENTS
1.1 MUSCLE PROTEIN METABOLISM
Place the aim of the study at the endo of this paragraph.
AUTHOR RESPONSE: The goal of this narrative review was to summarize available evidence surround the potential efficacy of plant proteins towards exercise outcomes. In light of your suggestion, we have stated this aim in section 1.3 after plant proteins has been introduced.
1.2 IMPORTANCE OF ADDED PROTEIN TO THE DIET
Line 53. A more appropriate title may be “The importance of added protein to optimize training adaptation”.
Why is there so much interest in using plant proteins to promote muscle mass and function?
AUTHOR RESPONSE: The title has been changed per your suggestion. We begin to answer the question you pose in the later sections. Section 1.1 was to highlight protein metabolism, 1.2 the need for more protein, and then 1.3 introduces plant proteins. Thus, the reasons for why plant proteins are of interest are detailed in 1.3.
1.3 THE CASE FOR PLANT PROTEINS
This paragraph is useless and a few concepts of interest may be easily included in para 1.4.
AUTHOR RESPONSE: Section 1.3 is needed to highlight to the reader why plant proteins are important. In considering the aim and title of the paper, it is our assessment this information is important.
1.4 QUALITY CONSIDERATIONS
I think that this paragraph needs to be made clearer and much more concise.
AUTHOR RESPONSE: Protein quality has been the central tenet for the last 20+ years why certain proteins were emphasized. For readers who do not understand how protein quality is assessed, this paragraph offers an overview of key details. We have reviewed the paragraph and removed what content we could, however, we feel all parts of this section are needed to help explain to the reader what key considerations exist surround different sources of protein.
The difference in protein quality between and within animal and plant proteins needs to be more adequately explained in terms of digestibility and amino acid composition.
AUTHOR RESPONSE: We agree. The first part of section 1.4 spends time outlining how animal and plant proteins compare in terms of different ways to assess amino acid composition. The second part outlines other important factors (digestibility and absorption kinetics) that differentiate animal and plant proteins.
It should be underlined that the authors focused on the evaluation of single protein sources (not of mixed diets).
AUTHOR RESPONSE: This is an excellent point. We have added a sentence in our section 9. Conclusions and Future Directions to make clear note of this for the reader.
PDCAAS and DIAAS take into consideration both digestibility and IAA composition. If I remember correctly, PDCAAS ranges between 1 and 0. DIAAS has been specifically proposed for evaluating the protein quality of specific food souces, also in order to define claims. Digestibility is much higher in plant protein isolates. Amino acid composition of soy protein should be reported.
AUTHOR RESPONSE: We have chosen not to highlight the composition of one protein source more than the other as this article is focused on helping the reader better understand the available literature involving plants. In addition, on several occasions was have referenced the Gorissen paper, which we feel is the best paper to highlight amino acid composition of different protein sources.
What are the plant proteins that have been used to optimize training adaptation? What are the plant proteins the authors are interested in? What is their protein quality? Comments on other plant proteins (legumes, cereals, etc.) could be misleading for the reader.
AUTHOR RESPONSE: We discussed all of the available studies involving different plant sources and exercise training adaptations in section 4. As such, we are not necessarily interested in any particular source of protein (whether its from plant or animals), and instead have written the paper to help the reader better understand the current state of literature surround the efficacy of various plant proteins in conjunction with exercise training.
Leucine content should be considered and treated in this paragraph.
AUTHOR RESPONSE: We agree and the last 3 sentences of the first paragraph of section 1.4 discuss the importance of leucine in terms of anabolic potential and protein quality.
Line 128-129 The statement is not completely true. A score below 1 just indicates that the amount of IAA/g protein provided by a given protein is lower than the maximum one. In several plant proteins, this is mainly due to a reduced digestibility.
AUTHOR RESPONSE: This sentence has been removed.
Line 135 Notably, the added values of DIAAS identifies which amino acids are limiting. Quite vague…
AUTHOR RESPONSE: Agreed and we feel it isn’t needed. So this sentence has been deleted.
- METHODS
This section should be completely rewritten according to standard section structure: PRISMA indications, selection of papers, researchers involved, quality of papers, etc.
AUTHOR RESPONSE: A non-systematic, non-meta, narrative approach was taken to prepare this review. To your point, we reviewed the PRISMA guidelines and added several more lines of content for readers to better understand why we wrote the review and what types of studies and outcomes were included in our paper. We have outlined conflicts of interests and have also outlined what authors completed what steps.
- ACUTE STUDIES
More than 1800 words to appraise eight papers. Too much.
AUTHOR RESPONSE: Author Response: We have reduced the length of this section with an effort to provide a more succinct overview of mechanistic properties and pertinent study findings. While there are only 8 studies that were included in this section, differences in acute anabolic properties of varying plant sources underpin the practical differences observed following long-term ingestion of each respective protein source. Therefore, a thorough description of these acute differences in anabolic properties is warranted. Furthermore, the topic of animal versus plant-proteins is a very popular point of discussion in the nutrition field with a large (and growing) body of evidence exploring the advantages and disadvantages of each. When one considers that many people are still discounting plant proteins because of the long withheld dogma that animal proteins are better, the need for more context and details is important.
Key messages are getting lost in too much text. Sometines, there is a wordy dscription of studies in a mechanistic sequence. The description of each paper needs to be shortened, avoiding the repetition of data provided in the Tables.
AUTHOR RESPONSE: Thank you for the feedback. The description of studies has now been modified to reduce the length of the section and provide more precise summaries of meaningful study findings.
It is not clear to the reader whether all the studies reported in Table 1 were actually focused on physical training.
AUTHOR RESPONSE: The column labeled “Exercise Program” summarizes any exercise protocol that was employed as part of the study protocol, if applicable. The N/A is used in instances where no exercise protocol was used. This has been added to the table legend for clarification.
Be more concise and focus on studies related to physical exercise.
AUTHOR RESPONSE: We have revised this section to be more concise with our study descriptions. However, we feel as though acute supplementation studies without exercise interventions, still provide value to the reader and context to the overall scope of the review article by summarizing the acute anabolic properties of each protein source. Therefore, we have kept the non-exercise studies in this section but noted these within the table for clarification.
Line 229. What does “net balance in protein levels” mean?
AUTHOR RESPONSE: Net whole-body protein balance is a way to describe the net result of protein synthesis vs. protein breakdown in response to protein ingestion with or without exercise as described by Park et al.
- PROLONGED STUDIES USING PLANT PROTEINS AND EXERCISE
Be more concise and make concepts more accessible and concrete.
AUTHOR RESPONSE: We have proofread this section and made changes to improve clarity. Additionally, we removed several sentences from the end of this section.
Key messages are getting lost in too much text. Sometines, there is a wordy description of studies in a mechanistic sequence. The description of each paper needs to be shortened, avoiding the repetition of data provided in the Tables.
AUTHOR RESPONSE: Thank you for bringing this to our attention. We have closely gone through this section and attempted to shorten where we can and also make our presentation of data more consistent.
- RECOVERY CONSIDERATIONS FOR PLANT PROTEIN SOURCES
Be more concise and make concepts more accessible and concrete.
AUTHOR RESPONSE: We have reviewed this section and removed content where possible. This section, however highlights designs and outcomes of different study approaches and different protein sources, thus, more context is needed to ensure the reader can appropriately understand the differences between each study.
- CONSIDERATIONS FOR OLDER ADULTS
This is another story…
AUTHOR RESPONSE: Appreciate your comment. We have closely reviewed this section and removed some content and added additional details to help reiterate the important considerations of this section relative to the greater topic.
The paragraph could be deleted.
AUTHOR RESPONSE: This paragraph is included and needed because it highlights the only two studies that have been published comparing some form of plant protein consumption in older adults relative to their ability to impact body composition, performance, and function changes, all extremely important outcomes for older individuals and the development of improved understanding for how to offset the negative changes seen with aging.
- INCREASING THE ANABOLIC POTENTIAL OF PLAT SOURCES
Once more, the content of the paragraph is interesting, but the text is not straightforward nor easily readable. Focus on exercise with brief general considerations.
AUTHOR RESPONSE: We appreciate your comment. This section is intended to highlight ways in which feeding and exercise strategies can be taken to potentially impact the anabolic response seen to consuming different types of plant proteins. We have gone through to provide various edits to improve the clarity and crispness of our messaging.
- Where is this paragraph?
AUTHOR RESPONSE: We are sorry as we are unsure of what you are referring to.
- CONCLUSIONS AND FUTURE DIRECTIONS
To be reconsidered based on previous paragrphs and made more persuasive.
AUTHOR RESPONSE: Thank you for your comment. We have reviewed this entire section and added important details. Moreover, this paragraph nor the entire paper is not written as a persuasive argument rather one that highlights the available data involving plant proteins relative to various acute and prolonged outcomes.
OTHERS
Use MPS for muscle protein synthesis, PDCAAS instead of PDCAAs and DIAAS instead of DIAAs.
AUTHOR RESPONSE: Thank you. This change has been made.
There are some mispellings.
AUTHOR RESPONSE: Thank you for your comment. The entire document has been proofread.
Reviewer 2 Report
The manuscript present an interesting and comprehensive review, taking into account the present interest worldwide in consuming more plant proteins, trend that is continuous growing and of present relevance.
This manuscript is well organized. Literature is up to date.
The topic is of great importance, plant-based proteins offer environmental and health benefits, and research increasingly includes them in study formulas.
But plant-based proteins are less digestible than animal proteins, due to the different structure of plant versus animal proteins. Besides, the availability of amino acids derived from plant proteins is lower than that of animal proteins. The metabolic fates of amino acids derived from plant and animal proteins are thus different, leading to metabolic differences in peripheral tissues like skeletal muscle. By improving the amino acid composition could also compensate for the lower anabolic potential of plant-based proteins, and in the same time increasing the quality of protein intake. - This issue shall be maybe a bit described in the manuscript. BUT adding more details may interfere with my colleague opinion as being already a long manuscript.
Several studies have been shown an inverse association between dietary protein intake and inflammation and oxidative stress scores, due to potential beneficial effect observed for higher total and animal protein intakes but was even more pronounced with higher plant protein intakes. - Therefore, it is obviously the importance of plant-based protein. An increase in plant-based protein intake could improve the ability of plant-based proteins to induce skeletal muscle mass gain and enhance their potential to support muscle mass maintenance.
Author Response
REVIEWER 2
Manuscript ID nutrients-1215584
Plant Proteins and Exercise: What Role Can Plant Proteins Have in Promoting Adaptations to Exercise?
Comments and Suggestions for Authors
The manuscript present an interesting and comprehensive review, taking into account the present interest worldwide in consuming more plant proteins, trend that is continuous growing and of present relevance.
AUTHOR RESPONSE: We appreciate the positive feedback.
This manuscript is well organized. Literature is up to date.
AUTHOR RESPONSE: Thank you for your positive comment.
The topic is of great importance, plant-based proteins offer environmental and health benefits, and research increasingly includes them in study formulas.
AUTHOR RESPONSE: We agree and this is why we felt a detailed review that highlights many aspect of the available studies would help inform and guide future research on the topic.
But plant-based proteins are less digestible than animal proteins, due to the different structure of plant versus animal proteins. Besides, the availability of amino acids derived from plant proteins is lower than that of animal proteins. The metabolic fates of amino acids derived from plant and animal proteins are thus different, leading to metabolic differences in peripheral tissues like skeletal muscle. By improving the amino acid composition could also compensate for the lower anabolic potential of plant-based proteins, and in the same time increasing the quality of protein intake. - This issue shall be maybe a bit described in the manuscript. BUT adding more details may interfere with my colleague opinion as being already a long manuscript.
AUTHOR RESPONSE: You bring up an excellent point and we have attempted to both shorten our paper but also add key details like you have mentioned. To this point, we have added various points to highlights the important details you mention. Thank you for your comment.
Several studies have been shown an inverse association between dietary protein intake and inflammation and oxidative stress scores, due to potential beneficial effect observed for higher total and animal protein intakes but was even more pronounced with higher plant protein intakes. - Therefore, it is obviously the importance of plant-based protein. An increase in plant-based protein intake could improve the ability of plant-based proteins to induce skeletal muscle mass gain and enhance their potential to support muscle mass maintenance.
AUTHOR RESPONSE: Thank you for your insight. We agree and hope that our review will effectively summarize key outcomes and parameters of completed studies to researchers can design more impactful and insightful studies in the future.
Round 2
Reviewer 1 Report
I appreciate the great efforts that the authors have made in response to my questions and concerns. Indeed, the manuscript is still too long and wordy. I think that the authors should reconsider their approach to process writing.
My major concern is the fact that the authors present a narrative review which has clearly the structure of a systematic review (e.g. with respect to tables). Why did not the authors perform a systematic review?
In my opinion, the authors should be encouraged to submit a more concise systematic review instead of a narrative review.